# Activation of the PI3K/AKT/mTOR Pathway in Cajal–Retzius Cells Leads to Their Survival and Increases Susceptibility to Kainate-Induced Seizures

**DOI:** 10.3390/ijms24065376

**Published:** 2023-03-11

**Authors:** Nasim Ramezanidoraki, Driss El Ouardi, Margaux Le, Stéphanie Moriceau, Mahboubeh Ahmadi, Dossi Elena, Danae Rolland, Philippe Bun, Gwenaëlle Le Pen, Guillaume Canaud, Nadia Bahi-Buisson, Nathalie Rouach, Rebecca Piskorowski, Alessandra Pierani, Pierre Billuart

**Affiliations:** 1Institute of Psychiatry and Neuroscience of Paris (IPNP), Institut National de la Santé et de la Recherche Médicale (INSERM) U1266, Université Paris Cité (UPC), 75014 Paris, France; 2Institut Imagine, Université de Paris Cité, 75015 Paris, France; 3GHU-Paris Psychiatrie et Neurosciences, Hôpital Sainte Anne, 75014 Paris, France; 4Platform for Neurobehavioral and Metabolism, Structure Fédérative de Recherche Necker, 26 INSERM US24/CNRS UAR 3633, 75015 Paris, France; 5Center for Interdisciplinary Research in Biology (CIRB), College de France, Centre National de Recherche Scientifique (CNRS), INSERM, Labex Memolife, Université Paris Sciences & Lettres (PSL), 75005 Paris, France; 6Institut Necker Enfants-Malades (INEM), INSERM U1151, Université Paris Cité, 75015 Paris, France

**Keywords:** development, Cajal–Retzius cells, neuronal survival, PI3K/AKT/mTOR pathway

## Abstract

Cajal–Retzius cells (CRs) are a class of transient neurons in the mammalian cortex that play a critical role in cortical development. Neocortical CRs undergo almost complete elimination in the first two postnatal weeks in rodents and the persistence of CRs during postnatal life has been detected in pathological conditions related to epilepsy. However, it is unclear whether their persistence is a cause or consequence of these diseases. To decipher the molecular mechanisms involved in CR death, we investigated the contribution of the PI3K/AKT/mTOR pathway as it plays a critical role in cell survival. We first showed that this pathway is less active in CRs after birth before massive cell death. We also explored the spatio-temporal activation of both AKT and mTOR pathways and reveal area-specific differences along both the rostro–caudal and medio–lateral axes. Next, using genetic approaches to maintain an active pathway in CRs, we found that the removal of either PTEN or TSC1, two negative regulators of the pathway, lead to differential CR survivals, with a stronger effect in the *Pten* model. Persistent cells in this latter mutant are still active. They express more Reelin and their persistence is associated with an increase in the duration of kainate-induced seizures in females. Altogether, we show that the decrease in PI3K/AKT/mTOR activity in CRs primes these cells to death by possibly repressing a survival pathway, with the mTORC1 branch contributing less to the phenotype.

## 1. Introduction

Cajal–Retzius cells (CRs) are transient excitatory pioneer neurons that migrate tangentially at the surface of the cortex during development and also populate the borders of the hippocampal fissure [1]. They come in multiple types depending on their origins. At least four sources at the border of the pallium have been characterized: the cortical hem, the pallial septum, the ventral pallium, and the thalamic eminences [2,3]. Besides their well-known function in cortical layering through the secretion of Reelin, they also control multiple steps of cortical development from neuronal proliferation to dendritogenesis and in functional area formation [1,4]. In rodents, neocortical CRs are almost fully eliminated in the two first postnatal weeks according to a neuronal cell-death program, which remains to be explored. Our previous studies have shown that BAX-mediated apoptosis is responsible for the elimination of a fraction of CRs, especially those derived from the septum [5]. In addition, this cell-death mechanism is dependent on neuronal activity, since blocking their excitatory inputs from inhibitory neurons [6] or hyperpolarizing the CRs [7] leads to their partial survival.

Cell death and survival are two Janus-faced components of cell life, and inhibiting cell survival leads to the same outcome. Amongst the most well-known pathways involved in cell survival is PI3K/AKT/mTOR, whose activation inhibits apoptosis, potentially leading to human cancer when overactivated in proliferating cells [8]. Class I PI3K catalytic subunits transduce signals from either receptor tyrosine kinase (RTKs) or G protein-coupled receptors (GPCRs) to AKT, which in turn phosphorylates numerous regulators to activate the mTORC1 complex. This pathway is negatively controlled at different levels by various proteins such as PTEN (phosphatase and tensin homolog), which reduces PIP3 levels, or by the TSC1/2 complex (tuberous sclerosis complex), which acts as GTPase activating proteins to inhibit the GTPase RHEB, the main activator of mTORC1 [9]. In addition to its phosphatase activity on PIP3 at the plasma membrane, PTEN also functions in different compartments such as the nucleus, the endoplasmic reticulum, and the mitochondria, where it regulates transcription and calcium release from the ER and structure, respectively [10].

During brain development, the activation of the pathway secondary to mutations in its components is responsible for various cortical malformations, spanning from hemimegalencephaly to focal cortical dysplasia associated with intractable pediatric epilepsy [11]. The severity of the phenotype correlates with the occurrence of the somatic mutation: the earlier the occurrence, the worse the phenotype [12].

In the mouse brain, the conditional activation of the pathway leads to different outcomes depending on whether it is eliminated in proliferating cells or neurons [13]. The deletion of *Pten* in neural progenitors enhances cell proliferation [14], whereas it causes hypertrophy in early post-mitotic neurons without altering specification and radial migration in the principal neurons of the neocortex [15]. The pathway is also essential for synapse and dendritic development [16] and participates in synaptic plasticity [17]. PTEN removal in hippocampal granule cells leads to focal seizures or generalized seizures depending on the degree of mosaicism [18].

During postnatal cortical development, active pyramidal neurons inhibit the PTEN expression in interneurons and protect them from cell death [19]. Here, we investigated the role of the PI3K/AKT/mTOR pathway in the cell death of CRs derived from the ΔNp73 lineage, which includes 80% of all CRs in the neocortex. We show that this pathway is physiologically inhibited at the postnatal stage in CRs before they start to die through apoptosis. We then sustained its activation using a genetic approach to prevent CRs from death. Both gain and loss-of-function approaches of PI3K or PTEN, respectively, led to CR survival, whereas the activation of the mTOR pathway by TSC1 removal had little effect on the cell numbers. As expected for the activation of this pathway, the morphology and the intrinsic properties of the survival CRs were altered, but these cells were still active and expressed more Reelin in an overall preserved cortex. PTEN loss-of-function in ΔNp73-derived CRs reduced Reelin N-terminal cleavage in the cortex of young animals, leading to an increase in the full-length fraction with no change in the mTOR activity in the pyramidal neurons underneath. Finally, the persistence of CRs in layer I of the cortex did not alter the general behavior of the *Pten* model, but increased its susceptibility to kainate-induced seizures.

## 2. Results

### 2.1. Spatiotemporal Regulation of the PI3K/AKT/mTOR Pathway Activity in ΔNp73-CRs

ΔNp73-CRs constitute approximately 80% of the total population of CRs in the neocortex, namely, Septum-CRs and Hem-CRs [20]. It has been shown that ΔNp73-CRs undergo BAX-mediated apoptosis in the two first postnatal weeks. However, the trigger of such apoptosis and the mechanisms regulating the death of other CR populations are still unknown [5]. The PI3K/AKT/mTOR pathway is one of the major cell-survival signaling cascades involved in the regulation of mitochondria-mediated apoptosis and it is also known to induce cellular senescence in non-replicative cells [21]. To decipher whether the activity of this pathway might regulate the death of ΔNp73-derived CRs in vivo, we first looked at the physiological activity of the PI3K/AKT/mTOR pathway at prenatal and postnatal stages. We used the ΔNp73^cre/+^ line [20,22] to label CRs via the conditional expression of tdTomato (tdT) as a Cre recombinase reporter (Rosa26^mT^). Brain sections were analyzed along both rostro–caudal (Figure 1A) and medio–lateral (Appendix A) axes at two different stages: the embryonic stage 17.5 (E17.5) and at postnatal day 1 (P1) before the beginning of massive cell death. The activity of the pathway was monitored by double immunofluorescence staining for pAKT (Ser473) and pS6 (Ser240/244), two well-known biomarkers of the pathway (Figure 1B).

The quantification of the total number of tdT-positive cells in the marginal zone along the rostro–medial axis showed a 40 % reduction in level 1 (Appendix A), corresponding to either altered migration more rostrally or precocious cell death in this level of the neocortex, whereas no change was observed in levels 2 and 3 (Figure 1C). As expected from the expansion of the cortex size between the E17.5 and P1 stages, CR densities along the rostral–caudal axis were significantly reduced by 20 to 40% according to the level (Figure 1D and Appendix A) (Appendix A).

Next, to monitor the activity of the pathway, two related markers, pATK and pS6, were used to distinguish between the activity of the PI3K/AKT and/or mTOR pathway, respectively (Figure 1B,E). The proportions of tdT cells negative for activity (pink bar in Figure 1E), positive for one or the other marker (gray and green bars in Figure 1E), or positive for both (dark orange bar in Figure 1E) were quantified. The rostro–caudal analysis of the brain at E17.5 showed that 60 to 70% of the total tdT cells were positive for one or two markers (Figure 1E), with no difference between the levels. At the P1 stage, there was an overall 40 to 50% reduction in the activities of the pathway at all different brain levels in comparison to E17.5 (cells negative for any activity, Appendix A)—more specifically, the proportion of CRs positive for two markers (cells positive for both markers, Appendix A) (Figure 1E). None of the single-labeled cells, pAKT or pS6 alone, showed a significant difference between the two stages along the rostro–caudal axis (cells positive for only one marker, Appendix A).

In contrast, the medio–lateral analysis of the brain sections at E17.5 showed a spatial difference in the activities of PDK1 or mTORC2 (pAKT) and S6 kinases (pS6) in CRs (Appendix A left panels). More CRs positive for p-AKT alone were present in the medial part of the neocortex (pAKT vs. tdT, Appendix A), whereas CRs positive for pS6 alone showed a mirrored pattern with an enrichment more laterally at all three brain levels (pS6 vs. tdT, Appendix A). Cells positive for both markers showed a differential pattern according to the brain level. At levels 1 and 3, they showed mirrored distributions with fewer in the lateral and medial regions, respectively (Appendix A left panels) (pS6 and pAKT vs. tdT, Appendix A), whereas at level 2, they followed more or less the same distribution as tdT-positive CRs (Appendix A, left panel). With the expansion of the cortex size, besides the reduced activation of the pathway (Figure 1E), the different patterns of phosphorylation observed at E17.5 tended to flatten at P1, but overall, they presented the same pattern of distributions with fewer statistical differences compared to tdT-positive cells (Appendix A, right panels) (Appendix A).

Altogether, the PI3K/AKT/mTOR pathway activity in ΔNp73-CRs presents a differential activation of AKT and mTOR pathways along the medio–lateral axis and a reduction in its overall activity from the prenatal to postnatal stages. This spatiotemporal control of the pathway activity is compatible with the hypothesis that the downregulation of a pathway involved in cell survival may contribute to CR death later on during the first two postnatal weeks.

### 2.2. Constitutive Activation of PI3K/AKT/mTOR Pathway Activity in ΔNp73-CRs Leads to Cell Survival

The perinatal period is a critical developmental time window for neurons as many of them are partially or almost eliminated [23]. To test whether the observed reduction in PI3K/AKT/mTOR pathway activity during this period is responsible for cell death in ΔNp73-CRs, we explored the consequences of maintaining the pathway permanently active in these cells. To do so, we used the conditional mouse model R26StopFloxp110* to constitutively express an activated version of the catalytic subunit of PI3K, the PIK3CA in CRs [24]. Upon the CRE recombinase-mediated deletion of the STOP cassette, the mutated p110alpha protein is expressed together with tdT reporter in ΔNp73-derived CRs (ΔNp73 > P110*). This mutant mouse model was first validated by monitoring the pathway activity at the P1 stage using pAKT and pS6 markers in the cortex (Appendix A, top panels). The quantification of the tdT cells showed an increase in the proportion of double-positive cells for the two markers by three to four times compared to controls at the three levels (Appendix A) (CR positive for both markers, control versus PIK3CA, Appendix A). None of the proportions for single labeling showed a significant difference between the two stages, suggesting that strong AKT activation also leads to the activation of mTOR. Thus, this genetic model allows the activation and maintenance of the activity of the pathway in CRs at a time when it should be physiologically downregulated.

Next, we determined whether the activation of PI3K interferes with the embryonic development of CRs since ΔNp73 is expressed from E11.5 in post-mitotic cells. We found that the cortical thickness was slightly increased in the future somatosensory brain area of level 2 (Appendix A) (Appendix A), but the tdT-positive CR densities (Appendix A) (Appendix A) did not differ between p110* and controls at the P1 stage in the rostro–caudal levels (control versus PIK3CA, Appendix A). Therefore, in this context, any difference in cell numbers after 2 weeks could only be attributed to cell persistence. At the P24 stage (Figure 2A), the CR densities were reduced from approximately 15,000 (Appendix A) (Appendix A) to 400 cells/mm^3^ (Figure 2C) (Appendix A) in the controls, as previously reported [5]. In contrast, the p110*-expressing animals (Figure 2A) showed a thicker layer I in the S1 barrel field (S1Bf) area of level 2 (Appendix A) together with a significant thee- to four-fold increase (1150 cells/mm^3^) in the tdT-positive cell density (Figure 2C) (Appendix A) compared to the controls at the same age at all three brain levels (Figure 2D) (control vs. PIK3CA, Appendix A). Since we observed some specific activation patterns of the pathway at E17.5 along the medio–lateral axis (Appendix A), we investigated the distribution of the ΔNp73-derived CRs in different cortical areas at P24 (Figure 2E). As expected from a gain-of-function experiment with strong pathway activation, almost all areas showed a significant increase in CR densities, with different increases according to the area (Figure 2F,G and Appendix A) (control vs. PIK3CA, Appendix A).

### 2.3. Conditional Inactivation of Tsc1 or Pten, Which Encode Two Negative Regulators of the PI3K/AKT/mTOR Pathway, Results in Specific Patterns of CR Survival

The results obtained from the expression of the p110* clearly demonstrate the postnatal survival capacity of the pathway in ΔNp73-derived CRs. However, since p110* activated the AKT/mTOR pathway everywhere in this cell population, the specific pattern of activation according to the brain regions or cortical areas observed in E17.5 embryos may have been altered. We thus turned to a loss-of-function approach to activate the pathway only in places where it is controlled and used the conditional inactivation (cKO) of either Tsc1 or *Pten*, which encode for two well-known negative regulators of the PI3K/AKT/mTOR pathway. Besides the mTOR pathway, these molecules, especially PTEN, are at the crossway of multiple other pathways [10]. We thus tested their respective contribution to the ΔNp73-derived CR survival by comparing both phenotypes. At P1, in these two models’ tdT CRs have different levels of activation according to the neocortex levels (Appendix A). The loss of TSC1 function increased the proportions of cells positive for pS6 at level 3, but not at levels 1 and 2, suggesting that the mTOR pathway is activated more in the caudal parts of the cerebral cortex. Regarding the *Pten* model, only level 3 showed a large increase in the proportion of double-positive CRs, whereas the proportion of pAKT single-labeled CRs was unexpectedly decreased at level 1 (Appendix A) (control and PTEN, Appendix A).

Next, we determined the consequences of the activation on the development of ΔNp73-derived CRs in both the *Tsc1* and *Pten* models. At the P1 stage, although the inactivation of Tsc1 did not alter the cortex size (Appendix A) or the densities of tdT-positive cells in the marginal zone of the cortex (Appendix A) (Appendix A), PTEN removal increased the cortical thickness in the S1Bf brain area (Appendix A) (Appendix A) and reduced the tdT-positive CR densities overall (Appendix A) (Appendix A). Thus, the embryonic development of ΔNp73-derived CRs is lightly altered in *Pten* cKO, but not in *Tsc1*.

We then quantified the CR densities at P24 in *Tsc1* and *Pten* cKOs (Figure 2C). At this stage, the pathway was still activated in both models, as revealed by the pS6 staining increase in the tdT CRs (Figure 2A). The loss of TSC1 function increased by almost double the number of tdT-positive CRs at all brain levels (control versus TSC1, Appendix A), whereas the loss of PTEN increased it three- to six-fold depending on the brain levels, similarly to the p110* model (Figure 2E,D) (control versus PTEN, Appendix A). A more detailed examination of the brain areas at each level (Figure 2E,D) shows that Pten cKO had a stronger phenotype than *Tsc1* since the densities were higher and affected more brain areas compared to *Tsc1* cKOs (TSC1 versus PTEN, Appendix A). From the comparison of the three levels (Figure 2B), both cKOs altered the medial cortical regions of the cortex, namely, the prefrontal cortex (ACA), retrosplenial–motor (RSP-MO), and retrosplenial (RSP) cortex, characterized by the highest densities of ΔNp73-derived CRs in the cortex (Appendix A and Figure 2F) (Appendix A). The areas of the intermediate section (level 2) were also more sensitive to the loss of either molecule compared to the rostro (level 1) and caudal (level 3) sections (Figure 2F,G and Appendix A).

In addition to these commonalities, the Pten cKOs presented significant increases in more lateral brain areas (Figure 2F,G and Appendix A) (Appendix A). When normalized to respective controls, the fold changes varied according to the areas, especially at levels 1 and 2, which showed an increase along the medio–lateral axis (Figure 2G and Appendix A) (Appendix A). This suggests that lateral CRs are more sensitive than the medial ones to the loss of PTEN, indicating that the endogenous pathway is likely to be more inhibited in the lateral cortex. This hypothesis is strongly supported by the differential pattern of pAKT staining observed at P1, especially at level 2 (Appendix A) (Appendix A). Finally, the *Tsc1* cKO did not present any specific brain area with an increase in CR density not also observed in the Pten cKO, suggesting that there is no brain area where the activation of the mTOR pathway would be uncoupled from the AKT pathway.

Altogether, ΔNp73-derived cells are differently sensitive to the loss of TSC1 or PTEN function depending on the rostro–caudal axis, with a stronger effect at level 2 and also on the medio–lateral axis with higher numbers of survival CRs in the medial cortex at all levels. Finally, ΔNp73-derived cells are more sensitive to *Pten* cKOs than *Tsc1*, suggesting that the activation of the mTOR pathway contributes little to the survival phenotype observed after either PI3K activation or PTEN loss of function.

### 2.4. Pten Inactivation in CRs Alters Their Morphology and Intrinsic Firing Properties

Alongside the increase in the CR number after the activation of the PI3K/AKT pathway, the Pten model clearly presents some obvious morphological alterations in the size and number of CR dendrites and/or axons at P24 (Figure 2A and Figure 3A,B) (Appendix A). These morphological changes resulting from polarity defects and neurite outgrowth have already been reported for other Pten conditional models where the AKT pathway has been activated [25,26]. Surprisingly, this phenotype in dendrites and/or axons was less penetrant after p110* expression than after PTEN loss of function (Figure 2A), which suggests partial compensation by endogenous PTEN in the P110* model or a PIP3-independent mechanism of dendritic elongation and branching in CRs regulated by PTEN. In addition, very few alterations to CR morphology were observed at the P1 stage in the Pten model (Appendix A), suggesting that the dendritic phenotype was acquired during the two first weeks and progressed with age, as previously reported [25].

We next investigated whether the surviving CRs in the Pten model at P24 were still active neurons. To this end, we prepared coronal slices of mutant and control animals and performed whole-cell current clamp recordings of tdT-positive cells in the somatosensory barrel S1cortex. All recorded cells were filled with biocytin to label individual CRs (Figure 3A). We observed that at P24, the CRs from the control animals were extremely difficult to locate, and the soma too small and membranes too fragile to achieve reliable electrophysiological recordings. Therefore, we recorded control cells at P10 before they died to enable a comparison with the Pten model [5]. We found that CRs from the Pten model had numerous intrinsic properties and physiological features consistent with active neurons (Figure 3D–F). They had more depolarized resting membrane potentials than the control (Figure 3D1; Appendix A), higher input resistance (Figure 3D2; Appendix A) and higher membrane capacitance (Figure 3D3; Appendix A), which is in agreement with their increase in neurite extensions and membranes. To examine the action potential firing properties of CRs in control and Pten mice, the cells were held at −75 mV and depolarizing current steps were injected to induce action potential firing (Figure 3D). From these experiments, we observed that Pten CRs fired significantly more action potentials per current step for current steps larger than 80 pA (Figure 3E1, Appendix A) and had larger action potential amplitudes (Figure 3E2, Appendix A) compared to the controls. Additionally, the AP threshold as well as the AP halfwidth were significantly reduced (Figure 3F1,F2, Appendix A) and lastly, the AP latency was longer for Pten CRs than the controls (Figure 3F3, Appendix A). These results for Pten CRs at P24 are consistent with the surviving cells being larger, electrically active, and capable of repetitive firing compared to control CRs, but maintaining their intrinsic CRs characteristics overall.

### 2.5. Increase in Reelin Expression Does Not Alter Cortical Layering in Pten Mutants or Neuronal Network Activity Patterns in S1 Somatosensory Cortex

As CRs are characterized by the expression of the Reelin protein, we tested whether the persistent CRs in our models continued to express it at P24. We found that they not only did, but also displayed an increase in signal intensity (Figure 2A). The quantification of Reelin intensities in Pten mutant CRs, which present the most extreme phenotypes, showed a significant increase in expression compared to the controls at P24 (Figure 3C) (Appendix A). Secreted Reelin appears as diffuse signal bands in both the cortex and hippocampus (Figure 4A). To investigate whether the observed increase in Reelin expression at P24 is also associated with more secretion in the Pten mutant, we analyzed a younger stage before CR cell death occurred in the controls to allow a direct comparison between the genotypes independently of the cell numbers. We selected four regions of interest (ROIs) devoid of cell bodies in layer I of the future S1 barrel field brain area and quantified at P1 the intensity of the Reelin signal in these ROIs (Figure 4A see yellow squares in the middle panel of the inset). Overall, the Reelin signal intensities in layer I of the Pten mutant brains were similar to the controls (Figure 4B and Appendix A). Full-length Reelin is secreted and cleaved at the N- and C-terminal sites by external metalloproteinases into five fragments, depending on the location and the stage [4]. To investigate the cleavage pattern of Reelin, we extracted total proteins from whole cortex at the P7–P8 stages and ran Western blot experiments. The expression of the full-length Reelin protein and also its cleaved fragments, NR-6 and NR-2 resulting from C-terminal and N-terminal cleavages, respectively, were analyzed in Pten and control mice (Figure 4D,E) using an anti-Reelin antibody recognizing an N-terminal epitope. Although the expression of full-length Reelin in the whole cortex was not different at this stage (Figure 4E and Appendix A), the fraction of cleaved Reelin was reduced by 15% in the Pten model (Figure 4F) (Appendix A)—more specifically, the N-terminal cleavage of Reelin leading to an increase in the full-length Reelin fraction in the cortex. This overall reduction in Reelin cleavage in the cortex at P7–P8 together with the increase in Reelin expression in ΔNp73-derived CRs at P24 had no impact on the general cortical organization of the Pten mutant brain, since it presented with similar thicknesses of the layers in the S1Bf at the P24 stage (Appendix A) (Appendix A).

Reelin activates the mTOR pathway in the surrounding cells through binding to its receptor, the phosphorylation of DAB, and the activation of PIK3/AKT/mTOR/S6 kinase to regulate the dendritic growth of targeted cells [27]. We therefore used pS6 staining to monitor the S6 kinase activity downstream of mTOR in the Pten mutant and control cortex at P1 (Figure 4A). Although the quantification of the signal intensity along the cortical thickness in the future S1 barrel field area (Figure 4A, yellow line) at P1 clearly showed some variations between the cortical layers, no significant difference could be observed between the genotypes (Figure 4C) (Appendix A). Therefore, the increased number of CRs together with the increase in Reelin expression and the reduction in its cleavage does not lead to a non-cell autonomous activation of the mTOR pathway in the cortex.

### 2.6. Persistence of ΔNp73-Derived CRs in Layer I Increases the Susceptibility of PTEN or p110* Females to Kainate-Induced Seizures

Somatic mutations in the PIK3/AKT/mTOR pathway in pyramidal neurons are responsible for various brain malformations and epilepsy depending on the mutation occurrence during brain development [11]. In addition, we recently reported that Bax inactivation in ΔNp73-derived CRs leads to their persistence in the cortex and hippocampus and increases the excitatory/inhibitory transmission ratio and the susceptibility to kainate-induced seizures [5,28]. To test whether the deletion of Pten also altered neuronal network activity patterns, we investigated the S1 cortex, a region where the CR densities and the Reelin expression were largely increased in the Pten model at P24. For this aim, we recorded population activity spontaneously generated in pro-bursting ACSF (Mg^2+^-free with 6 mM KCl (0Mg6K ACSF)) in S1 cortical slices from control and Pten mutant adult mice (P75) using the multi-electrode array (MEA) technique (Figure 4G). We found that S1 slices from both control and mutant mice mostly displayed bursting activity (n = 12 out of 14 control slices (85.7%) and n = 13 out of 15 mutant slices (87 %); Figure 4H–J), while only a minority of the slices developed seizures (n = 2 slices for both genotypes) in 0Mg6K ACSF (Appendix A). In addition, bursts recorded in control and mutant mice displayed a similar frequency and duration (Figure 4J) (Appendix A). These results thus indicate that Pten deletion in CR cells does not affect neuronal network activity in the S1 cortex.

We then used an in vivo approach to induce seizures by injecting kainate into adult mice in order to test the susceptibility to seizures. The general behavioral characterization of the Pten mutants animals did not display any clinical manifestations using the SHIRPA protocol [29] up to 80 days old, at which point they started to manifest cutaneous alterations (see behavioral section in the Materials and Methods section). More specific tests such as the open-field and the Y maze were also performed to monitor locomotor activity, and anxiety and working memory, respectively (Appendix A) (Appendix A). In agreement with the previous phenotypic characterization of Reelin-overexpressing mice, no significant difference was found according to gender or genotype in any of the tests [30]. Finally, we injected some of the previously analyzed mice with kainate (15 mg/kg IP) and monitored the latency, occurrence, and duration of the seizures over 120 *min* (Appendix A). Since the Pten model generally presents a larger variability in several readouts, we also used the P110* model to activate the AKT/mTOR pathway and compared the two approaches regarding their susceptibility to kainate-induced seizures. No difference in the latencies and the occurrences could be observed between the genotypes and sex (Appendix A, left and right panels, (Appendix A), but we noticed a consistent gender difference in the total duration of the seizures in both mutant mouse models (Appendix A, middle panels). Females of both genotypes, from p110* or Pten models, had longer seizures compared to their littermates (Appendix A, middle panels, blue or pink bars vs. gray bar) (Appendix A), suggesting that they remained frozen for longer during the few occurring seizures.

Altogether, the increased number of CRs following the activation of the PI3K/AKT/mTOR pathway did not alter general behavior, but increased the brain’s susceptibility to acute seizures in females.

## 3. Discussion

The PI3K/AKT/mTOR pathway is one of the major cell survival signaling cascades involved in the regulation of mitochondria-mediated apoptosis and has been shown, amongst others, to control interneuron cell death [19]. Here, we focused our study on a particular type of excitatory neuron, CRs, which populate the surface of the cortex during embryogenesis and die postnatally after orchestrating cortical development. We previously showed that Bax-mediated apoptosis and neuronal activity are involved in CR death [5,7]. By investigating the PI3K/AKT/mTOR activity around postnatal stages, just before massive cell death occurs in ΔNp73-derived CRs, we found that it is decreased, suggesting its participation in the control of CR cell death as well (Figure 1E). Besides this temporal regulation of its activity, we also highlighted a differential spatial activation of AKT and S6K along the medio–lateral axis at three rostro–caudal levels. Overall, AKT and mTOR activities are higher in the medial/dorsal and lateral cortex, respectively, at E17.5 and P1, suggesting that they are differentially regulated in the marginal zone above the future cortical areas (Appendix A). We then sustained the activation of the pathway in these cells in vivo using a genetic approach, and found that a fraction of them survived at P24 (Figure 2) in the cortex up to 110 days, whereas they should be almost eliminated by day 15. By conditionally targeting *Pten* and *Tsc1* in ΔNp73-derived CRs, we found that AKT activation contributes most to the survival CR phenotype compared to mTOR (Figure 2A). In addition, the increase in the CRs in the *Pten* model was higher in the lateral cortex than in the medio–dorsal (Figure 2G), which correlates with the lower AKT pathway activation described at E17.5 and P1 in this region (Appendix A). Altogether, the PI3K/AKT pathway is more inhibited in the lateral cortex compared to medio–lateral, which may confer to the lateral CRs’ greater susceptibility to cell death. The various timing of CR elimination along the medio–lateral axis may influence the development of the future brain areas as well. Since the secondary somatosensory auditory cortex presents with a 10-fold increase in CR at P24 in the *Pten* model (Figure 2G), it would be interesting in the future to challenge the animal behavioral response to various tests involving auditory stimulation (i.e., pre-pulse inhibition or fear conditioning test).

In this study, we also had the opportunity to test and compare two models of activation through either the expression of an activated PI3KCA mutant protein (p110*) or the deletion of PTEN. Both approaches should increase the PIP3 levels to activate AKT through PDK1 and one would have expected a phenocopy between the two mutants and even a stronger phenotype for the P110*, which has been shown to strongly activate AKT [31]. Although the number of persistent CRs is similar between the two mutants (Figure 2A,C), the *Pten* mutant clearly exhibits a large increase in dendrite and/or axonal extensions, which is rarely seen in the p110* model. One hypothesis may be the compensation of PIP3 overproduction in P110* mutant CRs by endogenous PTEN, but one would expect to also have such a compensation effect for cell survival and fewer CRs in P110* compared to the *Pten* model. Alternatively, PTEN may also have PIP3-independent functions [10]. Axodendritic outgrowth requires some reorganization of actin/microtubule cytoskeleton and, besides the known regulation of F-actin dynamics by phosphoinositides through Rho GTPases, PTEN directly regulates the stability of the microtubules through their detyrosination [32]. Apart from these morphological differences in CR projections, both mutant CRs present with an increase in the soma size together with an increase in pS6 staining. These two phenotypes are also observed in the *Tsc1* mutant CRs (Figure 2A) and are linked to mTORC1 activation and the increase in protein synthesis [9].

Surprisingly, whereas the loss of *Tsc1* in the cortex has been shown to increase the basal dendrite arborization of pyramidal neurons, a phenotype reversible upon rapamycin treatment during a critical period [33], only bipolar CRs were observed in *Tsc1* cKO (Figure 2A). This suggests that the activation of the mTORC1 pathway alone is not sufficient to alter the neuronal polarity and/or the axodendritic outgrowth in CRs, but it is probably necessary, since these morphological changes require an increase in protein synthesis [16].

Altogether, these results strongly suggest that PTEN regulates the growth of CR extensions by both AKT-dependent and -independent pathways. This drastic morphological increase in dendritic arborization is consistent with a large increase in membrane capacitance (Figure 3). We also observed several other changes consistent with a more excitable phenotype. This includes a decreased action potential threshold and input resistance, and increased number of action potentials at current steps higher than 60 pA. Previous reports have demonstrated altered hippocampal expression profiles of voltage-activated ion channels in PTEN knockout animals [34]. We postulate that similar changes in membrane excitability occurred in the CRs in our preparation.

Our results, together with the well-known effects of AKT activation on cell survival [35], strongly suggest that apoptosis is inhibited in our *Pten* model, which leads to an increase in CR densities in postnatal cortex. In theory, *Pten* and *Bax* inactivation should have similar secondary phenotypes linked to CR persistence. They have similar increases in cell densities, about five-fold. The *Bax* model presents with an increase in dendrites and dendritic spines in upper pyramidal neurons, which leads to an alteration of the excitatory to inhibitory balance and hyperexcitability together with an increase in susceptibility to kainate-induced seizures [28]. Although we did not investigate the morphological changes of these pyramidal cells in the *Pten* model, neuronal network activity is unaffected in the S1 barrel cortex. However, both *Bax* and *Pten* animals do present an increased susceptibility to seizures. Interestingly, both models showed an opposite gender differential effect, with the *Pten* model affecting females. While the role of testosterone in models of temporal lobe epilepsy has already been investigated [36], sex differences in the protein expression of the AKT/mTOR pathway have been shown in the hippocampus and cortex [37], and these differential expressions may explain the increased duration in the seizures of *Pten* females after kainate injection.

## 4. Materials and Methods

### 4.1. Animals

*ΔNp73^CreIRESGFP^(ΔNp73^Cre^)* [20] and *Ai9 line (Strain#* 007905) *ROSA26^loxP-stop-loxP-Tomato^(R26^mT^)* [38] transgenic mice were kept in a C57BL/6J background. *R26StopFlp110** mice (strain# 012343) were a gift from Dr. Guillaume Canaud (University Paris Cité) [24]. *Pten^loxP^* mice [39] and *Tsc1^loxP^* mice [40] were provided by Dr. Mario Pende (University Paris Cité). Animals were genotyped by PCR using specific primers (Appendix A). Different cohorts of control and mutant animals were produced according to the experiments and are summarized in Table 1.

All animals were handled in strict accordance with good animal practice as defined by national animal welfare bodies.

### 4.2. Tissue Preparation and Immunohistochemistry

The staging of animals was conducted considering midday of the vaginal plug formation as embryonic day 0.5 (E0.5) for the embryonic stage and the birth date as postnatal day 0 (P0) at the postnatal stage. Brain collection and fixation of both embryos and P1 animals were performed as previously described [2]. For juvenile mouse (P24 days after birth) brain sections, the animals were anesthetized by the intraperitoneal administration of a mixture of sedative (xylazine 10 mg/kg) and anesthetic (ketamine 180 mg/kg) substances and were intracardially perfused by 4% paraformaldehyde (PFA) in PBS 1× (*v*/*w*), pH 7.4 and post-fixed 2 h in 4% PFA at 4 °C. The brains were rinsed in PBS for 1 h and placed in 30% sucrose in PBS 1× (*v*/*w*) overnight for cryoprotection, and then embedded in OCT compound (Sakura). Embedded tissues were sectioned on a cryostat with a 20 µm thickness for embryonic and P1 stage and 50 µm for P24 brains. Three sections from rostral, intermediate, and caudal brain levels were chosen from the controls and mutants. Immunostaining on embryonic and P1 sections was performed as previously described [2]. The P24 brain sections were washed first in PBS 1× and then in 0.2% triton + PBS1X before immunostaining. Afterwards, the sections were permeabilized in PGT (PBS1× 0.25% triton 0.2% gelatin) for 1 h and then incubated with primary antibodies diluted in PGT at 4 °C overnight. The second day after washing, the sections were first incubated in PGT for 10 min and then were incubated in a secondary antibody diluted in PGT at room temperature for 1 h. Double immunostaining of two rabbit antibodies was carried out using a Zenon kit (Zenon Alexa Fluor 488 Rabbit IgG Labeling Kit, Z25302, Invitrogen by Thermo Fisher Scientific (Waltham, MA, USA)). The primary antibodies used for immunohistochemistry were: goat anti-Reelin antibody (R and D Systems (Minneapolis, MIN, USA) AF3820, 1:500), rabbit anti-phospho-AKT serine 473 antibody (Cell Signaling Technology (Danvers, MA, USA) 4060, 1:200), rabbit anti-phospho-S6 antibody (Ser 240/244 Cell Signaling Technology(Danvers, MA, USA) 5364, 1:1000), rat anti-ctip2 antibody (Abcam (Cambridge, UK) ab18465, 1:250), rabbit anti-cux1,2 antibody (Santa Cruz (Dallas, TX, USA) sc-13024, 1:250). The secondary antibodies used against the primary antibodies were: donkey anti-rabbit Alexa-488 (711-545-152, Jackson ImmunoResearch Laboratories(Cambridge, UK),1:500), donkey anti-goat Alexa-488 (A-11055, Molecular Probes(Eugene, OR, USA), 1:500), donkey anti-goat Cy5 (705-175-147, Jackson ImmunoResearch Laboratories (Cambridge, UK), 1:250), and donkey anti-rat Alexa-Cy5 (712-175-153, Jackson ImmunoResearch Laboratories(Cambridge, UK), 1:250). DAPI (D1306, ThermoFisher Scientific (Waltham, MA, USA), 1:1000) was used for nuclear staining. Sections were mounted using Vectashield (H-1000, Vector Labs(Burlingame, CA, USA)).

### 4.3. Image Acquisition and Analyses

Immunofluorescence images were acquired using a confocal Leica SP8 microscope (Leica Microsystems, Mannheim, Germany) at the E17.5 and P1 stages and a slide scanner Nanozoomer 2.0 (Hamamatsu) with a 20× objective for the quantification of CRs at P24. Confocal images were acquired using a 40× oil immersion objective, pinhole-sized “airy 2” and four z-stacks of 5 µm were defined in order to acquire sections of 20 µm thickness.

Then, the X and Y positions of each of the tdT-positive cells were recorded from a 150 µm depth straightened cortex using Fiji software [41]. For each section, the density of CRs (tdTomato + CRs/mm^3^) was calculated considering the thickness of the section, the neocortical length at P1 or the area lengths at P24 and the depth from the surface of the cortex.

Illustrated images with bigger magnification (Figure 2 and Appendix A) were acquired with a 40× and 3 digital zoom. 9 to 10 z-stacks with an optical section of 2 µm were acquired.

Illustrated images including biocytin-filled CRs (Figure 3) were acquired using a 63× oil immersion objective. Then, 20 to 60 z-stacks of 2 µm optical section were defined depending on the thickness of the sections and the size of the neurons. tdTomato + neurons were counted and analyzed for their position using the ImageJ software. Three sections from each biological replicate at the embryonic, P1, and P24 stages were analyzed corresponding to three different brain levels.

### 4.4. Analysis of Reelin and pS6 Fluorescent Intensities in Layer I and in Cortex

Using Fiji software [41], the fluorescent intensity signal of extracellular Reelin in layer I of the primary somatosensory cortex at the P1 stage was measured in four different regions of interest (ROIs) devoid of any cell bodies. From the same cortical area, pS6 fluorescent intensities were measured in a ROI rectangle divided into 10 bins and drawn perpendicular to the surface of the cortex from LI to L6 in the primary somatosensory cortex. In both quantifications, the mean intensities of the ROIs were corrected by subtracting from their value the mean intensity of an ROI outside the cortex where signals were minimal (from cerebral nuclei).

### 4.5. Western Blot Analysis

Cortices from the *Pten* and control mice (*n* = 4) were dissected from the brain at P7–P8 and snap-frozen in liquid nitrogen before extraction in RIPA buffer (ref: #R0278; Sigma-Aldrich (Saint-Louis, MI, USA) with protease (ref: #04693116001; Roche (Bâle, Switzerland) and phosphatase (ref: #5870; Cell Signaling (Danvers, MA, USA)) inhibitors. Following this, 25 µg of the total protein extracts were run on a 3–8% Tris–acetate gradient gel (NuPAGE^TM^, Ref: #EA03785BOX; Invitrogen (Waltham, MA, USA)) and transferred to nitrocellulose membrane (ref: #GE10600002, Amersham^TM^ Protan^TM^, Sigma-Aldrich (Saint-Louis, MI, USA)) before overnight incubation with primary antibodies (mouse anti-Reelin G10 (1:500; MAB5364; Millipore (Burlington, MA, USA), mouse anti-pan Cadherin (1:1000; C1821; Sigma-Aldrich (Saint-Louis, MI, USA)) in a Tris buffer (0.1M; pH:8) sodium (1.5M) Tween 20 (0.05%) (TBST) solution containing 5% fat-free milk. After 45 min of incubation with horseradish peroxidase conjugated mouse secondary antibodies (1/20.000, ref: # 115-035-008; Jackson ImmunoResearch (Cambridge, UK)) and three washes in TBST, the membranes were incubated in chemiluminescent substrate (Super Signal Pico Plus ref: #34580; Thermo Fisher Scientist (Waltham, MA, USA)) before exposition and signal acquisition under a digital camera (Chemidoc^TM^ Touch Imaging System; Bio-Rad (Hercules, CA, USA). Signal intensities were analyzed using Image Lab Software (V6.1; Bio-Rad, (Hercules, CA, USA)). The Reelin expression (sum of full-length, NR-6, and NR-2 intensities) was normalized to pan-Cadherin and its cleavage products to total Reelin (sum of NR-6 plus NR-2 normalized to sum of full-length, NR-6, and NR-2).

### 4.6. Neuronal Soma Size Measurement

To extract the cell volume, we considered the entire z-stack to extract ROIs using the machine learning-based WEKA tool implemented on Fiji/ImageJ2 [42]. Beforehand, we trained a WEKA classifier on fluorescent series of Td-tomato-loaded cells set using the following features: Gaussian blur, Derivatives, Structure, Minimum, Maximum, and Median. Classification was performed using the FastRandom Forest algorithm. Upon extraction of 3D ROIs, we recorded the volume using the 3D ROI manager tool developed by Ollion and colleagues [43].

### 4.7. Patch-Cell Recording

Slice preparation: Coronal slices were prepared from *Pten* transgenic animals between P20 and P24. At these ages, the animals were anaesthetized with ketamine (100 mg/kg), xylazine (7 mg/kg), and isoflurane, and perfused transcardially with a sucrose-based cutting solution containing the following (in mM): sucrose 110, KCl 2.5, NaH_2_PO_4_ 1.25, NaHCO_3_ 30, HEPES 20, glucose 25, thiourea 2, Na-ascorbate 5, Na-Pyruvate 3, CaCl_2_ 0.5, and MgCl_2_ 10. Coronal slices were prepared from the control animals at P10, as CR cells were non-viable for electrophysiological recording after this age. At P10, the animals did not undergo cardiac perfusion. For both *Pten* and controls, the brains were rapidly removed, placed upright, and cut into 300 μm thick coronal slices (Leica Biosystems Vibratomes, VT1200S, (Wetzlar, Germany)) in the cutting solution at 4 °C. The slices were transferred to an immersed-type chamber and maintained in artificial cerebral spinal fluid (ACSF) containing the following (in mM): NaCl 125, KCl 2.5, NaH_2_PO_4_ 1.25, NaHCO_3_ 26, glucose 10, Na-pyruvate 2, CaCl_2_ 2, and MgCl_2_ 1. The slices were incubated at 32 °C for approximately 10 min and then maintained at room temperature for at least 45 min. Prior to recording, the slices were transferred to a recording chamber perfused with ACSF at 3 mL/min at 30 °C.

Electrophysiological recordings: CR cells were located using the fluorescence tdTomato tag with a 535 nm LED and long-pass mCherry filter set in an Olympus BX51 microscope (Olympus Corporation, Tokyo, Japan). A recording pipette with a resistance between 2 and 5 MΩ with positive pressure was inserted into layer I of the Barrel cortex. Whole-cell recordings were performed with intracellular solutions containing the following (in mM): K-methyl sulfonate 135, KCl 5, EGTA-KOH 0.1, HEPES 10, NaCl 2, MgATP 5, Na_2_GTP 0.4, Na_2_-phosphocreatine 10, and biocytin (4 mg/mL). Series resistance was <20 MΩ and was monitored throughout the recordings. Data were discarded if the series resistance changed more than 10% during the experiment. The bridge balance was measured every 20 sec and compensated with internal circuitry and monitored throughout the experiments. The liquid junction potential was not corrected. Data were obtained using a Multiclamp 700B amplifier and digitized using a Digidata 1440 ADDA board. Data were sampled at 10 kHz. Axon™ pCLAMP^®^ 9 electrophysiology data acquisition and analysis software (Molecular Devices, (San Jose, CA, USA)) was used for data acquisition. Action potential firing properties were measured with a series of one-second-long depolarizing current steps.

Immunochemistry and cell identification: Post hoc confirmation of all cells was performed by labeling intracellular biocytin. Following overnight incubation in 4% PFA in PBS, the slices were permeabilized with 0.2% Triton in PBS and blocked for 48 h with 3% goat serum (50062Z, by Thermo Fisher Scientific (Waltham, MA, USA)) in PBS with 0.2% Triton. Streptavidin and Alexa Fluor™ 546 conjugate (S11225, by Thermo Fisher Scientific (Waltham, MA, USA), 1:300) incubation was carried out in block solution for 4 h at room temperature. The slices were mounted in ProLong™ Diamond Antifade Mountant (P36970, Invitrogen by Thermo Fisher Scientific, (Waltham, MA, USA)), resulting in a partial clarification of tissue following incubation at room temperature for 24 h. Images were acquired.

Data analysis: Electrophysiological recordings were analyzed using custom-written macros with IGORpro (WaveMetrics Inc. (Lake Oswego, OR, USA) and Axograph software (Axograph, RRID:SCR_014284). Action potentials (APs) were detected automatically by threshold crossing with Axograph (Axograph RRID:SCR_014284) followed by visual inspection. R_M_ measurements were calculated from linear fits of the voltage responses to current step injections of 10 pA. Action potential firing properties were measured on a positive current step series starting at 20 pA with increasing 10 pA current steps. The threshold was measured at the first AP at rheobase. The AP full-width at half-maximal amplitude (AP width) and after-hyperpolarizing potential (AHP) amplitude were measured for the first AP at threshold. The latency of the first AP and number of APs was measured with increasing current steps over rheobase. The instantaneous firing frequency and amplitude of the first five APs were measured at current injection steps 1.5 times over rheobase. Sag potentials were measured with a protocol injecting a current as necessary to maintain a V_M_ of −70 mV with a one-second-long negative current step resulting in a membrane hyperpolarization of −100 mV. The sag was calculated as the difference between the peak and steady-state voltage during the 1 s hyperpolarization.

### 4.8. Multi-Electrode Array (MEA) Recordings

All experiments were performed on adult (P75) female mice. The mice were sacrificed and the brain removed. Coronal slices (400 μm) of S1 cortex were cut at low speed (0.04 mm/s) and at a vibration frequency of 70 Hz in ice-cold oxygenated artificial cerebrospinal fluid (ACSF) supplemented with sucrose (in mM: 87 NaCl, 2.5 KCl, 2.5 CaCl_2_, 7 MgCl_2_, 1 NaH_2_PO_4_, 25 NaHCO_3_, and 10 glucose, saturated with 95% O_2_ and 5% CO_2_). The slices were maintained at 32 °C in a storage chamber containing standard ACSF (in mM: 119 NaCl, 2.5 KCl, 2.5 CaCl_2_, 1.3 MgSO_4_, 1 NaH_2_PO_4_, 26.2 NaHCO_3_, and 11 glucose, saturated with 95% O_2_ and 5% CO_2_) for 20 min, and then stored for at least 1 h before recording in a pro-epileptic (0 mM Mg^2+^, 6 mM K^+^; 0Mg6K) ACSF. S1 cortical slices were transferred to planar MEA Petri dishes (200-30 ITO electrodes organized in a 12 × 12 matrix, with internal reference, 30 μm diameter and 200 μm inter-electrode distance; Multichannel Systems, Kusterdingen, Germany), and kept in place using a small platinum anchor. The slices were continuously perfused at a rate of 2 mL/min with 0Mg6K ACSF during the recordings. Images of the S1 cortical slices on MEAs were acquired with a video microscope table (MEA-VMT1; Multichannel Systems, Germany) through MEA Monitor software (Multichannel Systems, Kusterdingen, Germany) to identify the location of the electrodes on the cortex and to select the electrodes of interest. Data were sampled at 10 kHz and network activity was recorded at 32 °C by MEA2100-120 system (bandwidth 1–3000 Hz, gain 5×, Multichannel Systems, Kusterdingen, Germany) through the MC Rack 4.5.1 software (Multichannel Systems, Kusterdingen, Germany).

Multi-electrode (MEA) data analysis: Bursting raw data were analyzed with MC Rack (Multichannel Systems, Kusterdingen, Germany). The detection of bursts was performed using the “Spike Sorter” algorithm, which sets a threshold based on multiples of standard deviation of the noise (five-fold) calculated over the first 500 ms of recording free of electrical activity. A five-fold standard deviation threshold was used to automatically detect each event, which could be modified in real-time by the operator using a visual check if needed. Bursts were arbitrarily defined as discharges shorter than 5 s in duration. Typically, bursts were characterized by fast voltage oscillations followed by slow oscillations or negative shifts. To analyze seizure activity, the data were exported to Neuroexplorer (Nex Technologies, Colorado Springs, CO, USA). Paroxysmal events were identified as discharges lasting more than 5 s; successive paroxysmal discharges were considered separate events based on their waveform and on the presence of a minimum (>10 s) interval of silent or bursting activity between them [44].

### 4.9. Behavior

Three independent cohorts were generated to produce a total of 15 to 20 mice per genotype (control and *Pten* model) and per sex (males and females). Due to ΔNp73 expression in the epidermal and hair follicle stem cells [45], the *Pten* models start to develop skin accumulation under the paws and curly fur after 12 weeks that preclude later investigations for ethical considerations. Each cohort was submitted successively to an adapted SHIRPA test to clinically evaluate each animal at 8 weeks old, followed by the rearing activity (at 9 weeks old), the open field (at 10 weeks old), and the Y-Maze tests (at 11 weeks old) according to the International Mouse Phenotyping Resource of Standardised Screens (IMPReSS) recommendations (accessed on the 10 January 2020, https://www.mousephenotype.org/impress/index). Finally, 12-weeks-old mice were processed for kainate injection to test the susceptibility to seizures before euthanasia.

#### 4.9.1. Horizontal Activity

Spontaneous horizontal locomotor activity was assessed using the open field test (OFT). To do so, unacclimated mice were moved to the monitor area and immediately placed individually in the center of the motor activity boxes (40 cm L, 40 cm W, 40 cm H) placed on an infrared floor under ~50 lux of light measured in the center of the arena, for a 120 min period. The total distance traveled by the animal (horizontal activity) and the time spent in the center (25 cm × 25 cm) of the arena were recorded by an infrared camera system connected to the Videotrack v2.6 Automated Behavioural Analysis software (ViewPoint, Lyon, France). The results are indicated as distance traveled in centimeters.

#### 4.9.2. Vertical Activity

Spontaneous locomotor activity was also evaluated with an infrared beam system fitted to rectangular cages (35 × 25cm) (ActiMot, TSE Systems GmbH, Berlin, Germany, version V5.1.9). Movement of the mice resulted in the interruption of the infrared beams. Each interruption was automatically counted, and the coordinates of the animal were determined in three dimensions (X–Y–Z axis). Each unacclimated animal was placed in the center of the cage and the number of beam breaks recorded during a 1-h period. We examined and analyzed the vertical (Z beam breaks) and horizontal (X + Y beam breaks) activities directly from the data collected by the system software.

#### 4.9.3. Y-Maze

Short-term memory was assessed by spontaneous alternation behavior in the Y-maze task. Experiments were carried out using standard Y-maze apparatus made of three identical black plastic arms (60 cm arms; walls 10 cm) set at 120° angles from each other. Different visual cues (geometrical forms) were placed on the wall of two arms of the maze and were kept constant during the experiments. Mice were placed at the end of one arm of the Y-maze and the sequence of arm entries were recorded over 10 min. The introduction arm was randomly assigned for each animal. An arm visit was recorded when a mouse moved all four paws into the arm. An alternation was defined as consecutive entries into all three arms (e.g., A, B, C or A, C, B) and was automatically recorded using Videotrack v2.6 Automated Behavioural Analysis (ViewPoint, Lyon, France). The number of maximum alternations was the total number of arm entries minus two and the percentage of alternations was calculated as the ratio of actual to maximum alternations multiplied by 100: (actual alternations/maximum alternations) × 100. Persevering behavior was defined as subjects making significantly fewer alternations than would be expected by chance (50%).

#### 4.9.4. Kainate Injection and Susceptibility to Seizures

Animals at 12 weeks old were injected with a single intraperitoneal dose of kainate (#0222/10, 10 mg, Bio-Techne (Minneapolis, MIN, USA) at 15 mg/kg and recorded for 2 h by Videotrack v2.6 Automated Behavioural Analysis (ViewPoint, Lyon, France) software. The latency, duration, and the number of seizures and their severity were recorded by two users blind to the animals.

### 4.10. Statistical Analyses

The results are reported as mean ± SD (histological and behavioral analyses) or SEM (slice and cell recording) in the figures and supplementary tables. The statistical significance of the linear variables was assessed using paired or unpaired Student’s t test, multiple t test with inequal variance, repeated-measures ANOVA followed by post hoc tests where appropriate, and a non-parametric Mann–Whitney U test when the distribution was not normal. Normality was tested with the Jarque–Bera test or Shapiro–Wilk test. Prism8 software (GraphPad, Boston, MA, USA) was used to test for outliers, normality, statistical differences, and for graphical representations of histograms and plots. Each test is detailed in the supplementary tables and in the figure legends.

## 5. Conclusions

We have identified the PIK3/AKT pathway as a key player in the survival of ΔNp73-derived CRs, which is downregulated soon after birth, priming these cells to death after exposure to other stressors (i.e., excitotoxicity, ROS production, DNA damage). Future investigations will determine whether other sources of CRs share the same protective pathway as ΔNp73-derived CRs and which neuronal growth factors control the pathway.

## Figures and Tables

**Figure 1 ijms-24-05376-f001:**
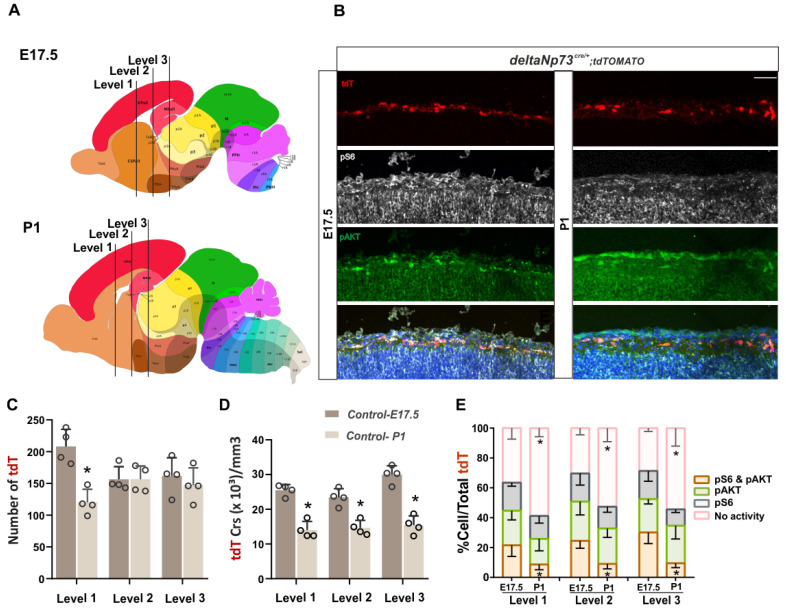
Reduction of the PI3K/AKT/mTOR pathway activity from E17.5 to P1. (**A**) Sagittal representation of brains at E17.5 and P1 (adopted from Allen brain). Black lines highlight three different levels, which were analyzed along the rostro–caudal axis. (**B**) Merged and single-channel confocal images of E17.5 and P1 control brains stained for pS6 Ser 240/244 (gray), pAKT Ser 473 (green), and DAPI (blue). tdTomato (red) traces of ΔNp73-derived CRs. (**C**) Number of tdT-positive cells at the neocortex levels 1, 2, and 3 (*n* = 4 for E17.5 and P1). (**D**) CR density (CRs/mm3) at the neocortex levels 1, 2, and 3 (*n* = 4 for E17.5 and P1). (**E**) Proportion of CRs positive for one or two markers or negative over total tdT cells (*n* = 4 for E17.5 and P1). Mann–Whitney U test (**C**–**E**); data are represented as mean ± SD; * *p* < 0.05. Scale bars: 50 µm.

**Figure 2 ijms-24-05376-f002:**
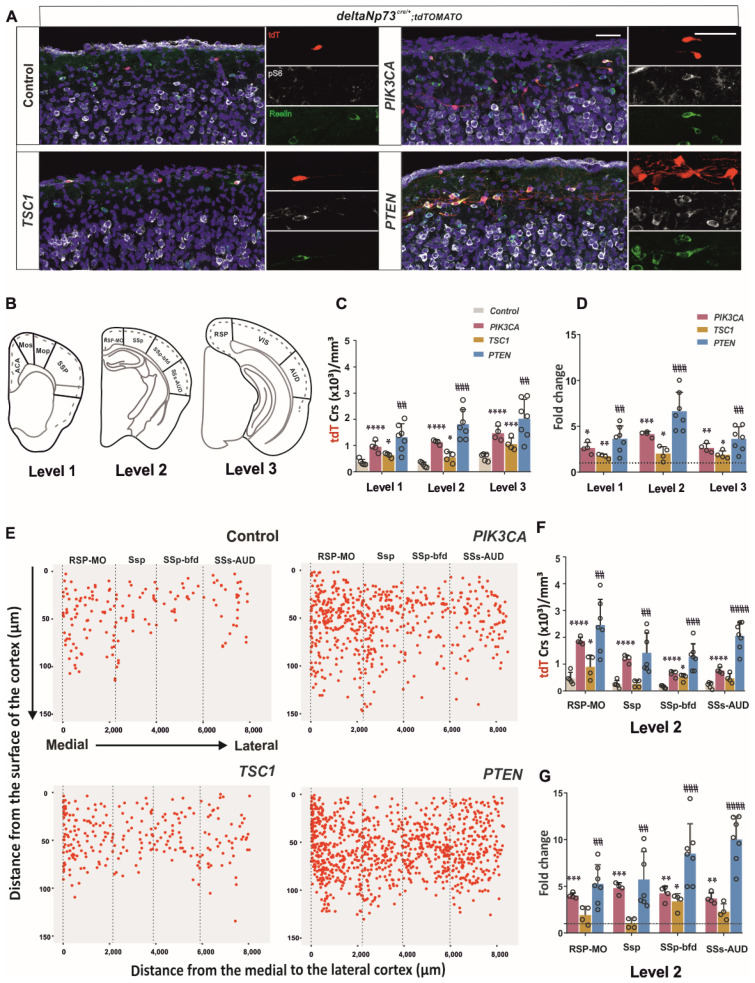
Sustained postnatal activation of the PI3K/AKT/mTOR pathway leads to the persistence of CRs at later stages. (**A**) Merged and single-channel confocal images of P24 control and mutant cortices in S1 barrel field stained for pS6 Ser 240/244 (gray), Reelin (green), and DAPI (blue). ΔNp73-derived CRs in tdTomato (red). The three single-channel images on the right are zoomed in from the left merged images. (**B**) Schematic representation of the analyzed medio–lateral axis along the three rostro–caudal levels. (**C**) CR density (CRs/mm^3^) in the rostro–caudal axis of the neocortex levels 1, 2, and 3 (*n* = 4 for control and mutants). (**D**) Fold changes in CR densities at the three levels (*n* = 4 for control and mutants). (**E**) Scatter plot of CR positions along the medio–lateral axis at level 2. (**F**) CR density (CRs/mm^3^) in the medio–lateral axis of the neocortex level 2. (**G**) Fold changes in CR densities in the medio–lateral axis of the neocortex level 2. Two-way ANOVA followed by a Sidak’s multiple comparisons test for control vs. PI3K and TSC1; multiple *t*-test for control vs. PTEN. (**C**,**F**) One sample *t*-test. (**D**,**G**) Data are represented as mean ± SD; * *p* < 0.05; ** *p* < 0.005; *** *p* < 0.0005; **** *p* < 0.0001. ## *p* < 0.005; ### *p* < 0.0005; #### *p* < 0.0001. Scale bar: 50 µm.

**Figure 3 ijms-24-05376-f003:**
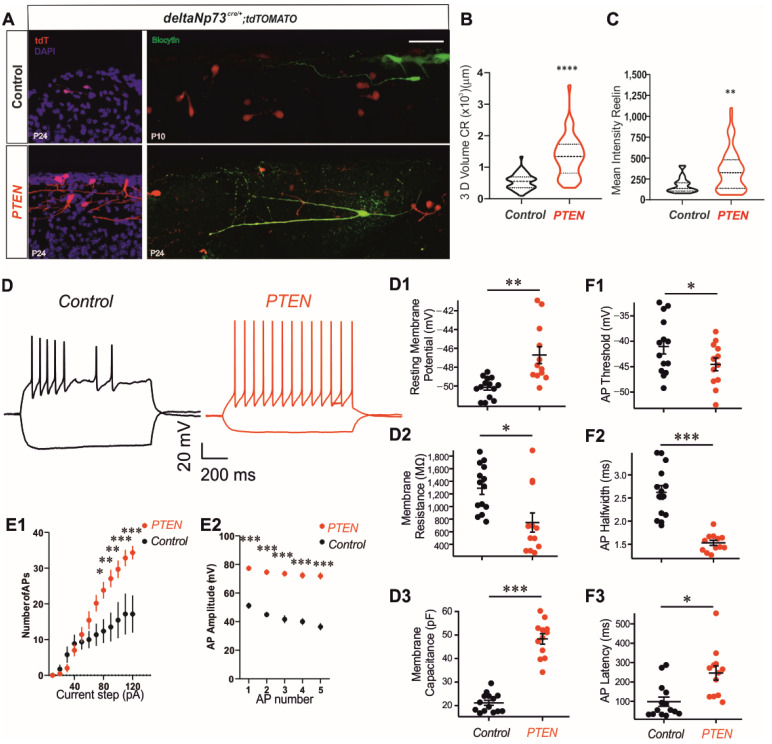
Inactivation of PTEN in ΔNp73-derived CRs alters their morphology and electrophysiological intrinsic properties and increases Reelin expression and AP amplitudes. (**A**) Left: merged-channel confocal images of P24 control and *Pten* brains with DAPI (blue) and tdTomato (ΔNp73-derived CRs). (**A**) Right: merged-channel confocal images of control (P10) and *Pten* (P24) brains after biocytin injection (green), tdTomato (ΔNp73-derived CRs). (**B**) Three-dimensional volume of tdTomato-positive CRs (µm^3^) at P24. (**C**) Mean fluorescent intensity of Reelin staining within CRs positive for tdTomato at P24. (**B**,**C**) Quantification from images in Figure 2A at P24 (control and *Pten*), Mann–Whitney U test; data are represented as mean ± SD. (**D**) Representative traces of current clamp recordings of CR cells recorded from control (black) and *Pten* (red)-model mice. Cells were held at −75 mV; a hyperpolarizing and a depolarizing current step are shown. Quantification of intrinsic properties of CR cells from control and *Pten* mice: (**D1**) resting membrane potential, (**D2**) membrane resistance, and (**D3**) membrane capacitance. (**E**,**F**) Action potential firing properties of CR cells from control and *Pten* mice: (**E1**) number of action potentials in one-second long depolarization as a function of current injection, (**E2**) action potential amplitude for the first five action potentials, (**F1**) action potential threshold, (**F2**) action potential halfwidth, and (**F3**) action potential latency. (**D**–**F**) Averages with error bars denoting SEM. * *p* < 0.05; ** *p* < 0.001; *** *p* < 0.0001. Scale bars: 50 µm.

**Figure 4 ijms-24-05376-f004:**
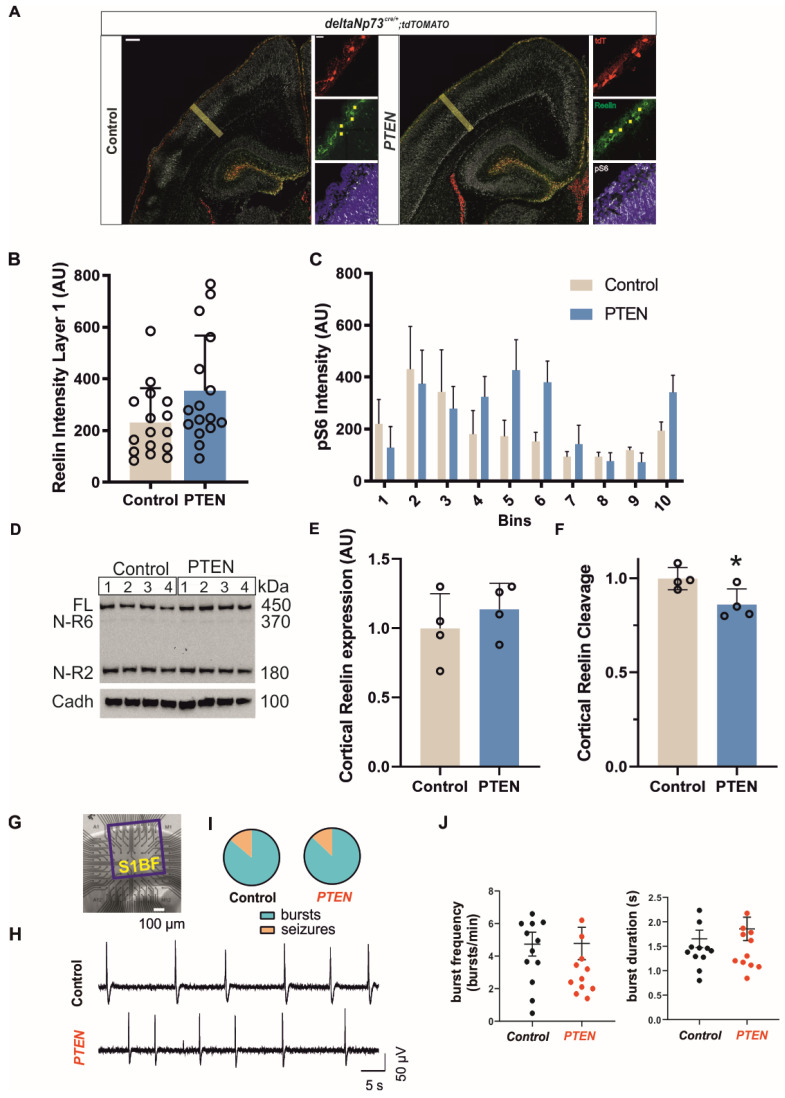
Inactivation of PTEN in ΔNp73-derived CRs reduces Reelin cleavage in cortex without altering mTOR activation or neuronal network activity in the S1 cortex. (**A**) Merged and single-channel confocal images of control and *Pten* mutant brains at P1 stage stained for pS6 Ser 240/244 (gray), Reelin (green), and DAPI (blue). ΔNp73-derived CRs in tdTomato (red). Scale bar: 180 µm. The three single-channel images on the right are zoomed in from the left merged images. Scale bar: 20 µm. (**B**) Quantification of the extracellular Reelin intensities determined from four ROIs in layer I (yellow squares in the Reelin zoom panels, A) (Mann–Whitney U test, *n* = 4). (**C**) Quantification of pS6 intensities determined from 10 bins along the thickness of the future S1 brain area (thick yellow line in panel A) (two-way ANOVA, *n* = 4). (**D**) Western blot analysis of Reelin expression and cleavage using G10 antibody against its amino-terminal end (top panel) and normalized to the ubiquitous pan-Cadherin expression in four controls and *Pten* mutants at P7–P8 (bottom panel). (**E**) Quantification of Reelin expression from (**D**) normalized to pan-Cadherin and to controls (t-test, *n* = 4). (**F**) Quantification of Reelin cleavage (Sum(N-R6;N-R2)) from (**D**) normalized to total Reelin (Sum(FL; N-R6;N-R2)) and to controls (* *p* < 0.05, *t*-test, *n* = 4). (**G**) Image of a cortical mouse slice placed on a multi-electrode array (MEA) dish. The S1B cortex is highlighted by the red square. Scale bar: 100 µm. (**H**) Representative MEA recordings of bursting activity in 0Mg-6K ACSF in control (top) and PTEN mutant (bottom) mice. Scale bars: 50 µV, 5 s. (**I**) Percentage of slices with bursting (blue) or seizure (orange) activity in control and PTEN mutant mice (*n* = 14 and 15 slices for control and PTEN mutant mice, respectively, from four mice for both genotypes). (**J**) Quantification of burst frequency and duration in control (*n* = 13 slices) and PTEN mutant (*n* = 14 slices) mice. Data are represented as mean ± SEM.

**Table 1 ijms-24-05376-t001:** Genetic backgrounds of control and mutant animals according to the experiments.

**Histology**		**Behavior**	
**Control**	**Mutant**	**Control**	**Mutant**
ΔNp73^Cre/+^ > R26^mT/+^	ΔNp73^Cre/+^ > R26^mT/+^; *Pten*^lox/lox^	ΔNp73^+/+^ > *Pten*^lox/lox^ or ΔNp73^Cre/+^ > R26^mT/+^	ΔNp73^Cre/+^ > *Pten*^lox/lox^
ΔNp73^Cre/+^ > R26^mT/+^	ΔNp73^Cre/+^ > R26^mT/+^; *Tsc1*^-/lox^	-	-
ΔNp73^Cre/+^ > R26^mT/+^	ΔNp73^Cre/+^ > R26p110*^lox/mT^	ΔNp73^Cre/+^ > R26^+/+^ or ΔNp73^+/+^ > R26p110*^lox/+^ or ΔNp73^+/+^ > R26^+/+^	ΔNp73^Cre/+^ > R26p110*^lox/+^
**Patch-Cell Recording**		**Multi-Electrode Array**	
**Control**	**Mutant**	**Control**	**Mutant**
ΔNp73^Cre/+^ > R26^mT/+^	ΔNp73^Cre/+^ > R26^mT/+^; *Pten*^lox/lox^	ΔNp73^Cre/+^ > R26^mT/+^	ΔNp73^Cre/+^ > R26^mT/+^; *Pten*^lox/lox^

## Data Availability

Not applicable.

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
