# Peer review of "Activation of the PI3K/AKT/mTOR Pathway in Cajal–Retzius Cells Leads to Their Survival and Increases Susceptibility to Kainate-Induced Seizures"

_ijms, 2023, doi:10.3390/ijms24065376_

Round 1

Reviewer 1 Report

The paper from Ramezanidoraki et al. investigates the role of the PI3K/AKT/mTOR pathway in the survival of Cajal-Retzius cells, a transient population of cortical neurons. The removal of PTEN or TSC1 affects the survival of CR cells, with PTEN having a stronger effect. Activation of the pathway leads to changes in the morphology and intrinsic properties of CR cells and to CR cells releasing more Reelin. This increase in Reelin has a non cell-autonomous effect in the mTOR pathway in pyramidal cells. However, no alterations in the neuronal network, sensitivity to kainate or mouse behavior were observed.

While the paper addresses an important question (how is the death of CR cells regulated), and several interesting results are reported, some of the results are not properly presented and some conclusions not properly supported. The authors relay a bit too often in simple literal observation to decide if something has or not an effect (see for example Pt 7).

Major points for revision:

1.       The authors put a lot of emphasis on the changes in Reelin secretion upon PTEN removal. This is included in the title, but strangely, not in the abstract. The main concern here is the way the authors decided to quantify Reelin. First of all, it is unclear how this was done. It is to be assumed it was done via immunohistochemistry, though nothing is reported in the Methods section (just some info in Legend of Figure 3). The comparison was done at P24, if understood correctly, though it is expected there will be a higher signal at that age as in the control most of the CR cells are gone. A comparison should be done also at P5-P10, when CR cells density might be more similar, to properly claim an increase in the release of Reelin. In addition, where was it quantified? In the CR cells? Around? Are the somata of CR cells excluded? Quantification of immunohistochemical signals is highly biased on the staining procedures. It is unclear if all the sections were processed exactly at the same time with exactly the same solutions prepared exactly the same day. A real quantification of Reelin levels should use biochemical methods (e.g. Western Blot). A microdissection of Ly1 would allow for a quantification of Reelin, and since Reelin is cleaved extracellularly, the comparison of the full-size protein Vs the cleaved isoforms would strengthen the conclusions (production VS secretion). A better understanding on the release of Reelin could also better explain the difference to the Bax model.

2.       The comparison of P24 CR cells in the PTEN group and of P10 CR cells in the control group is less than ideal. I do understand the difficulty of patching CR cells at P24 in the control group, but the authors could have used P10 for the PTEN group too. Indeed a comparison between the early and late stage in the PTEN group could increase the understanding of the reasons on the changes of CR Cells (e.g. are the changes only happening after a certain developmtal time?)

3.       Similarly, the authors say the PTEN manipulation is affecting the morphology of CR cells. While a nice image is included in 3A, it is unclear why the authors are not reconstructing the cells they patched (they have biocytin in the intracellular solution after all) and properly quantifying the changes in morphology. It seems the only measure is actually the soma size (Fig.3B), and, once again, P10 Ctrl Vs P24 PTEN are compared, in a less than ideal way.

4.       The authors mention an increase in pS6 staining in the cortex (Fig.4A). However, this is based on pure observation, and no quantification is present (same issue as in Pt1)

5.       What could the different activation of the pathway(s) in different regions mean? Lines 227-228 say no correlation with activation and cell survival, however, in the discussion (line 424-27) the authors say there is a correlation between the pathway activation and the cell survival in the PTEN mice. The authors should elaborate better as it is a bit confusing.

6.       Some of the results could be better discussed. For example, what could be an explanation for the drastic difference in ephys properties of CR cells in PTEN mice? Some speculation on possible causes could be nice.

Minor points for revision:

7.       The description of the Results section has not been deleted (see lines 93-95). Similarly for appendixes (lines 800 and onward)

8.       No staining for pAKT in Figure S2 or in other figures in any of the manipulations (just in Fig1 in control animals)

9.       Line 175 claims no difference in cortical size Fig. S2A, but there is no quantification. Similar line 211. Though in later stages the authors quantify cortical thickness in at least one of their manipulations, so authors could refer to this data already in line 211’s paragraph.

10.   Color of bar graphs in Fig 2 doesn’t match the legend. No light pink, it looks dark yellow. The colors in Fig1E were also partially missing or mixed in one of the figure versions I had available, so I encourage the authors to recheck all the colors in all the figures.

11.   Authors should refer to Table 5.3 in line 293, not only to Fig.4B

12.   Line 577: typo in Ctip2 (CTIPI2)

13.   Figure 4C is never referred to in the main text.

14.   CR cells do survive way more than two weeks postnatally in the hippocampus. While the authors in the majority of the case refer specifically to neocortical CR cells, sometimes (intro and abstract) they do not specify. A brief statement to distinguish the two populations could be added.

Reviewer 2 Report

The article in my opinion is exhaustive and experiments are well planned. The article is accepted in the present form.

Author Response

We would like to thanks the reviewer for his positive evaluation of the Manuscript.

Round 2

Reviewer 1 Report

I would like to thank the authors for addressing all the comments. In the present form the papers presents stronger conclusions.